# A low-cost, open-source centrifuge adaptor for separating large volume clinical blood samples

Md Ehtashamul Haque[1], Linda Marriott[1], Noman Naeem[1], Taygan Henry[1], Alvaro J. Conde[1,2], Maïwenn Kersaudy-Kerhoas[1,3]*

1 Institute of Biological Chemistry, Biophysics and Bioengineering, School of Engineering and Physical Sciences, Heriot-Watt University, Edinburgh, United Kingdom, 2 Micronit B.V., Enschede, Netherlands, 3 Infection Medicine, College of Medicine and Veterinary Medicine, University of Edinburgh, Edinburgh, United Kingdom

* m.kersaudy-kerhoas@hw.ac.uk

## Abstract

Blood plasma separation is a prerequisite in numerous biomedical assays involving low abundance plasma-borne biomarkers and thus is the fundamental step before many bioanalytical steps. High-capacity refrigerated centrifuges, which have the advantage of handling large volumes of blood samples, are widely utilized, but they are bulky, non-transportable, and prohibitively expensive for low-resource settings, with prices starting at $1,500. On the other hand, there are low-cost commercial and open-source micro-centrifuges available, but they are incapable of handling typical clinical amounts of blood samples (2-10mL). There is currently no low-cost CE marked centrifuge that can process large volumes of clinical blood samples on the market. As a solution, we customised the rotor of a commercially available low-cost micro-centrifuge (~$125) using 3D printing to enable centrifugation of large clinical blood samples in resource poor-settings. Our custom adaptor ($15) can hold two 9 mL S-Monovette tubes and achieve the same separation performance (yield, cell count, hemolysis, albumin levels) as the control benchtop refrigerated centrifuge, and even outperformed the control in platelet separation by at least four times. This low-cost open-source centrifugation system capable of processing clinical blood tubes could be valuable to low-resource settings where centrifugation is required immediately after blood withdrawal for further testing.

## 1. Introduction

A centrifuge is one of the most frequently used instruments in laboratory diagnostic and molecular biology laboratories, where it is employed to extract particles having different densities from a variety of mediums, using centrifugal forces. The primary uses of a centrifuge in a laboratory include the separation of plasma from whole blood for immunoassays or hematocrit analysis [1], the separation of pathogens and parasites in biological fluids [2], and DNA extraction preparation steps [3].

**Data Availability Statement:** All files are available in Figshare https://figshare.com/articles/dataset/Centrifuge_characterisation_data/16762444.

**Funding:** M.K.K. received funding from the UK Engineering and Physical Science council EP/R00398X/1 https://epsrc.ukri.org/ The funders had no role in study design, data collection and analysis, decision to publish, or preparation of the manuscript.

**Competing interests:** The authors have declared that no competing interests exist.

In particular, blood plasma separation is an essential, and often primary step in numerous biomedical assays involving low abundance target molecules, such as cell-free nucleic acids. Conventionally, this separation process is performed using high-capacity refrigerated centrifuges which are capable of dealing with a large volume (>5–50 mL) of blood samples. These laboratory centrifuges are bulky (120 kg), usually work with 20–40 cm diameter rotors holding 20–100 sample vials, thus occupy a sizeable space in the laboratory space. Moreover, these large capacity centrifuges are expensive, (capital expenditure starts from $1,500) and have high operational costs [4]. There are commercially available micro-centrifuges that can be cost-effective whose price starts from $120, however their drawbacks are that firstly they cannot handle volume blood samples above 1.5 mL and secondly, they cannot handle routinely used clinical tubes such as S-Monovettes (Sarstedt, Germany) which introduces further preparation steps before the centrifugation. Many researchers have adopted microfluidic blood plasma separation to enable low-cost plasma separation; however, these systems still have yield and purity issues for extremely low abundance biomarker detection, and the low throughput (10 mL/h) of the developed high-purity ones makes them inadequate for practical high-volume blood processing applications. In many clinical applications, such as Apolipoprotein E and HLA detection [5, 6], early cancer detection [7, 8], and liquid biopsy [9, 10], large volumes of plasma (>4 mL) are required. Hence, there is a clear demand on the market for a centrifugation system that will at the same time be safe, low-cost, able to handle large volumes as well as work with routinely used clinical blood tubes. A system like this might be useful for mobile laboratories or low-resource environments when centrifugation is necessary immediately after blood collection for additional testing.

To reduce the cost of centrifugation, a number of so-called "frugal" solutions have emerged, as part of the open-source and open-science movement [11–13]. Centrifuges using hand powered rotary mechanism such as fidget spinners [14, 15], egg beaters [16], paper toy [17], hand crank torch lights [18], salad spinners [19], centrifuges involving electric motor such as USB fans [20, 21], and Dremel tools [22] have been proposed, amongst other solutions. However, all of these devices are only able to handle low volume samples typically from a few microliters to 2 mL. Only a few solutions have been proposed to meet the large volume centrifugation requirement. Patel *et al.* constructed a portable, low-cost 3D printed microcentrifuge with a DC motor that can accommodate special 4 mL glass tubes, which necessitates additional blood handling phases [23]. Sule *et al.* also designed a 3D printed hand-powered centrifuge for high volume centrifugation that costs only $27 in total [1]. Because it lacks a protective cover, the gadget might not suitable for use with biological samples. Plasma separation needs at least 10 minutes of centrifugation, which will be difficult to achieve using a hand-powered microcentrifuge. Most crucially, none of the abovementioned low or high-volume systems can be used with clinical blood collection tubes (such as S-Monovette) straight after blood removal, resulting in further sample preparation steps. Fig 1A shows the processing volume of available commercial and open-source academic microcentrifuges against their price. Fig 1B illustrates the size difference between the blood handling tubes commonly used in microcentrifuges and the clinical S-Monovette collection tube.

In this work, we propose a novel approach to significantly reduce the cost of S-Monovette centrifugation by customising the rotor of a commercially available microcentrifuge (SciSpin MINI Microfuge, model: SQ-6050) using low-cost additive manufacturing (3D printing) (Fig 1C). 3D printing has emerged in recent years as a convenient method for the development of cost-effective and open-source scientific and diagnostic tools [12, 24–26]. Here we describe the design and implementation of the 3D-printed rotor adaptor. Using a combination of modelling and experimental validation, we report on the effects of different engineering parameters (size, mass, aerodynamic drag force) and leverage this knowledge to optimise the design and

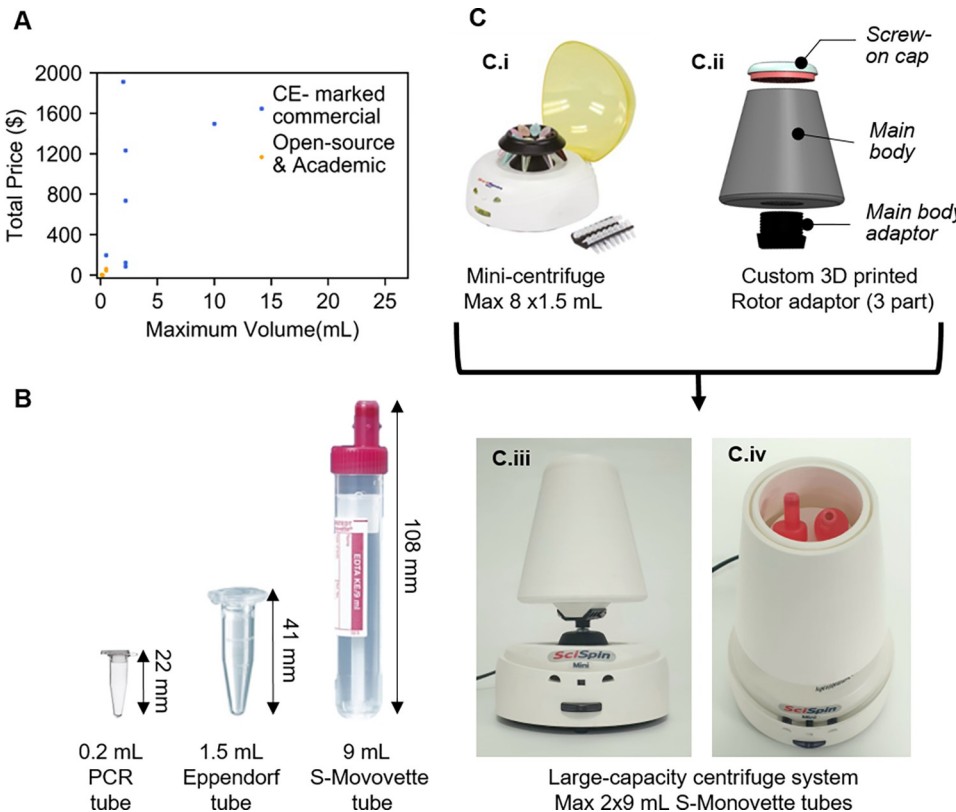

**Fig 1. (A)** Sample processing volume vs price of available commercial and open source academic microcentrifuges, **(B)** Illustration of the different sized tubes used with the available microcentrifuges along with S-Monovette tube used in this project, **(C)** Adaptor concept: **C.i)** Commercial SciSpin MINI Microfuge, model: SQ-6050 with its original rotors **C.ii)** CAD schematic of the designed three-part rotor adaptor **C.iii)** Final 3D printed rotor adaptor mounted on the commercial microcentrifuge base **C.iv)** Top cover removed from the adaptor, showing inside part of the rotor adaptor that holds two standard 9mL S-Monovette tubes.

speed of the rotor adaptor. We demonstrate that the optimised 3D-printed rotor adaptor achieved same speeds as the original rotor (7,000 rpm) despite the larger load and using plasma yield, blood cell counts, cfDNA and albumin levels we provide a full biological characterisation of the optimised devices.

## 2. Material and methods

### 2.1 Theoretical background

Centrifugation is a way to increase the gravitational field magnitude by spinning a sample around an axis which creates a relative centrifugal force (RCF, also called the "g" force) capable of pulling cells and other particles to the furthest position from the centre of rotation. The RCF is proportional to the square of the rotor speed and the radial distance and can be calculated using the following equation:

$$RCF = 1.118 \times 10^{-5} \times \bar{r} \times \omega^2 \qquad (1)$$

Where $\bar{r}$ is the average radial distance of the sample in the tube (in cm) and $\omega$ is the rotor speed (in Revolutions Per Minute, RPM). The schematics of the tubes and dimensions are shown in Fig 2A.

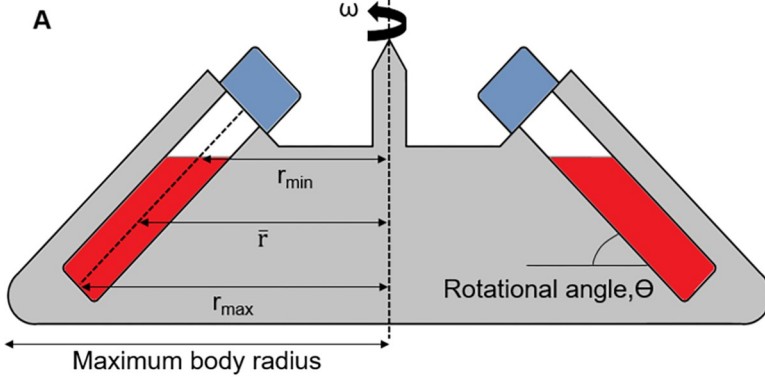

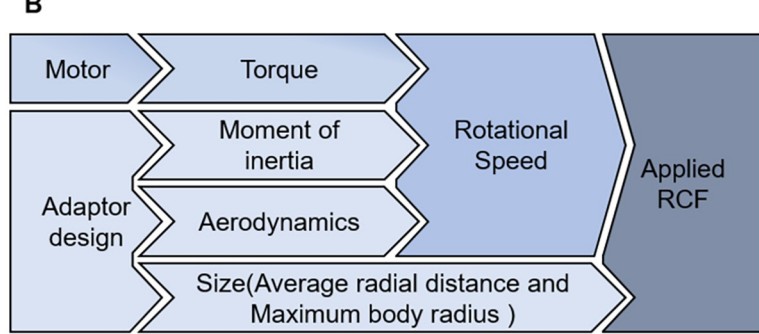

**Fig 2. (A)** Schematic illustration of the tubes on a fixed-angle adaptor and the radial dimensions **(B)** The influential parameters on the applied RCF on the sample in the tube mounted on the adaptor.

Although it seems from Eq (1) that the RCF should increase with an increment in speed and the distance of the sample from the axis, other parameters must be considered to obtain the highest RCF. These parameters include the moment of inertia, torque, and aerodynamics (Fig 2B). As the original rotor is going to be replaced by the customised one, to meet this additional load demand, the torque of the motor will increase, which will force the speed to go down because of the inverse relationship between the torque and speed. As the RCF increases with the square of the rotor speed, halving the speed will result in a 4-fold decrement in RCF which will result in poor separation efficiency. The additional mass and larger size of the rotor adaptor to accommodate the S-Monovette tubes lead to a decrease in the speed because of the larger moment of inertia and higher aerodynamic drag force. Therefore, although the increment of radial distance increases the RCF, it can also decrease the rotational speed because of the increased moment of inertia and aerodynamic drag force. In this trade-off between speed and radial distance, the speed has been given priority because of its squared relationship with the RCF and every effort has been made to keep the design as small and light as possible. In addition to these discussed parameters, the rotational angle also plays a vital role in the separation process where most compact pellet after the separation usually forms with a higher rotational angle and therefore 45° fixed angled rotors are most in commercial centrifuges that uses fixed-angle rotors. It should be noted that, among the two types of rotors, fixed-angle and swing-bucket rotors, the former one has many advantages like lower exposure to stress, higher RCF and not having any moving hanging parts; hence, adopted in our project. The effect of different angled rotors has been studied in different designs which will be discussed in the next sections.

## 2.2 Centrifuge hardware

The commercial microcentrifuge (SciSpin MINI Microfuge, model: SQ-6050) used in this project is widely available on mainstream purchasing platforms. To the best of our knowledge, this is the lowest CE-marked cost microcentrifuge on the market at the time of writing. It also has a higher rotational speed (7000 RPM), is lightweight and is more compact compared to other available devices. The full technical specifications of the microcentrifuge are provided in (S1 Table in S1 File). A commercial benchtop refrigerated centrifuge (Allegra X-12R, Beckman Coulter) with a swinging bucket, (Modular Disk Adapters for Tubes (SX4750)) was used as the benchmark. The benchmark centrifuge was always run with an RCF of 3273×g (3750 RPM). The RCF for the benchmark was chosen as the maximum speed allowed on a swinging bucket configuration and close to the average first spin values used in cfDNA studies [27]. The maximum speed for the commercial Allegra centrifuge is respectively 3750 and 10,200 RPM for swinging bucket and fixed angle, (equivalent RCF respectively 3720 and 11,400 ×g) The second spin protocol applied in plasma quality measurements was 12,000×g RCF for 10 minutes with a commercial high-speed microcentrifuge (5417 R, Eppendorf), also based on cfDNA studies [27].

## 2.3 3D printed adaptor fabrication

All 3D printed rotor adaptors presented here were designed using 3D modelling CAD (Solid-Works, 2018) and fabricated using Fused Deposition Modelling (FDM) technology. The Anycubic i3 Mega (Anycubic, Shenzhen, China) 3D printer was employed for printing. Polylactic Acid (PLA) (Verbatim 1.75mm clear PLA, brand) was used to print the rotor. The technical characteristic of the printer is provided in (S2 Table in S1 File). The objects were sliced with Ultimaker Cura 4.4 [28] using the standard settings summarized in (S3 Table in S1 File). All the design files are provided in.stl format in the online repository FigShare (https://doi.org/10.6084/m9.figshare.16762444.v2).

## 2.4 Simulation of critical speed

Any rotating system tends to vibrate in the absence of a driving force at certain frequencies called natural frequencies. When the frequency of the rotational speed matches with the natural frequencies of the system, there will be a resonance and the system will vibrate at that frequency. This natural frequency matching speed is commonly known as the critical speed of the system. The vibrations of the rotating system can impose high shear stress on the blood cells and could potentially damage the cells, leading to hemolysis, the destruction of red blood cells, and erroneous analytical results [29]. Therefore, it is important to minimize the critical speed to ensure the minimum shear stress on blood cells. Industrial centrifuge systems operate below and above critical speeds and the critical speeds are controlled via damping in rotor shaft connection [30]. In this project, the critical speed of different designs was predicted in-silico using Ansys workbench modelling (2021 R1, student edition) with the purpose of minimising the model critical speed. During the simulation, modal analysis has been selected from the available analysis systems. The designed 3D model was then imported into the analysis using the geometry tab. The remaining configuration was then completed in the analysis' model tab. PLA material was assigned in this step and the threads present in the original design were removed to facilitate the meshing. Fine meshing was chosen which was the highest possible meshing setting available with the Ansys version used in this study. Later the connection point between the designed rotor and the centrifuge base was selected as a fixed support. To run the modal analysis under loaded condition (rotational velocity) and calculate the critical speed using Campbell diagram, the Coriolis effect, the Campbell diagram and the damped

condition were turned on, and the number of points was set to 15, allowing us to do the modal analysis for fifteen different speeds. Finally, from the environment tab, a rotating velocity was added to the analysis, and fifteen rotational speeds were employed, ranging from 0–7000 RPM with 500 RPM intervals. Following the computation, a Campbell diagram was created under the solution, displaying the design's critical speeds, with the one lower than 7000 RPM chosen as the model critical speed for the further analysis.

### 2.5 Deflection measurement

The deflections of each 3D printed adaptor were measured from video recordings using a code written in the Python OpenCV module. Three video recordings of each adaptor were taken, with a duration of 4 minutes (At 0 sec, Motor mains and video start; at 2 minutes, Motor mains stop; at 4 minutes, video stop). To reach their respective maximum speed, all of the designs take 30–50 seconds, depending on their angular acceleration. The motor mains were switched off at 2 minutes to guarantee the designs ran for at least 1 minute at their maximum speed. Finally, the designs reach zero speed around 50 seconds to 1.5 minutes, depending on the load torque. Therefore, the video was cut off after 4 minutes. Prior to the deflection measurement, sizes of the video files were reduced using an online video converter (https://ezgif.com/). In the first module of the python code, the edge position of the adaptor was detected and transformed in pixel units for each frame of the video. Then these detected positions were compared with the initial reference and deflections in the pixel unit were incorporated into a final deflection matrix. Finally, the values of the deflection matrix were saved to an MS Excel file along with their corresponding pixel numbers. The code saved the screenshot of the processed video images from which a pixel per mm value was calculated from a known distance. This calculated pixel per mm value was then compared to the saved deflections values in the MS Excel file which provided the absolute deflection in mm for each design. The Python code is provided as S1 Code, along with an example video of Design C in S1 Video. Video recordings of all devices are available from the FigShare repository https://doi.org/10.6084/m9.figshare.16762444.v2

### 2.6 Sample material

Human blood samples were obtained under local ethical approval from the Scottish National Blood Transfusion Service (contract #18~06) and according to the Declaration of Helsinki. Samples were kept refrigerated (2–8° degrees) before their use. Blood samples were ordered from the same group (O positive) and pooled. Upon arrival, they were mixed gently to save Red Blood cells (RBCs) from getting damaged via excessive shear stress. Prior to each experiment, 9 mL blood samples (from the same pool) were poured into 9 mL S-Monovette opened syringes using S1 Pipet Filler (Thermo Scientific) which enabled fatigue-free pipetting. The larger diameter of the pipette tip (~0.9 mm) ensured lower stress exerted during blood aspiration.

### 2.7 ImageJ analysis on yield calculation

Prior to the experiments, all the S-Monovette tubes were marked to indicate the area required for a 1 mL volume of plasma. After each experiment, photographs of the tubes in a fixed custom set-up were captured with a mobile camera (Samsung Galaxy Note9). Thereafter, the captured images were processed using ImageJ software to measure the yield of the separated plasma. During the analysis, the previously marked area was selected as the scale. The plasma volume separated after centrifugation was calculated from the position of the plasma limit. Finally, the yield was calculated by comparing this separated plasma volume with the total

available plasma in the sample, known from the hematocrit (Hct) measurement (See section 2.8).

## 2.8 Blood cell counts

RBCs and platelets count, hematocrit (Hct), and hemoglobin (Hgb) of all the pre-and post-centrifuged blood and plasma samples were measured by a hematology analyzer (Sysmex XP-300, Sysmex Corporation, Japan). These measurements were used to compare the separation efficiency and purity of different designs.

## 2.9 Characterisation of hemolysis via spectrophotometry

A centrifuge operates at a high centrifugal force and may exert a shear rate on the RBCs resulting in hemolysis. The reference for hemolysis rate during storage provided by the American Society for Clinical Pathology is 2% or less [31], the Council of Europe guidelines recommend not to exceed 0.8% [32] and the US FDA 1%. While these values provide a useful guide to interpret the effect of centrifugation on blood samples, it should be noted that they refer to hemolysis in blood and blood components intended for transfusion or for further manufacture, not in-vitro diagnostics. During hemolysis, RBCs release their Hgb content on the sample. Free Hgb measurements in plasma provide an estimation of the overall hemolysis of a sample. In order to evaluate the hemolysis generated during the centrifugation with different devices, Hgb concentration on the plasma was measured using the spectrophotometric Cripps method at 560, 576 and 592 nm wavelengths [33]. The percentage of hemolysis was estimated with Eq 2 [34]:

$$Hemolysis\ Percentage(\%) = \frac{(100 - HCT) \times Free\ HGb}{Total\ Hgb} \tag{2}$$

where Hct and Total Hgb represents the total hematocrit and total Hgb content of the initial blood sample and free Hgb is the estimation of Hgb concentration using the Cripps method in an undiluted plasma sample. In this method, background absorption from other proteins, such as bilirubin, is automatically mitigated by the fractional absorbance between 576nm and at 560nm and 592nm wavelengths. In order to quantify the absolute free Hgb level in our samples, a standard curve was obtained by diluting human Hgb powder (Sigma-Aldrich, USA) in human plasma (Cambridge Bioscience, UK) to make samples of 1, 0.5, 0.1, 0.05, 0.01, and 0.005 mg/dL. The absorbance of the plasma samples extracted after centrifugation of different devices along with the standard samples made from human plasma was measured with the 96 well plate reader (POLARstar Omega, BMG Labtech).

## 2.10 cfDNA extraction

To assess cfDNA levels in several designs and controls, total cfDNA was extracted from 3 mL of separated plasma using the QIAamp Circulating Nucleic Acid kit (QIAGEN) following manufacturer instructions. Extracted cfDNA samples were frozen until use. Real-time quantitative PCR was performed using 2x Power SYBR® Green PCR Master Mix (Thermo Fisher Scientific) to amplify 90 bp target with LINE primers (final concentration 200 nM): forward `5'-TGC CGC AAT AAA CAT ACG TG -3'` and reverse `5'-GAC CCA GCC ATC CCA TTA C-3'` [35]. A standard curve was created using a series of 5 dilutions of Human Genomic DNA. Thermal cycling conditions involved a 10-minute cycle at 95˚C followed by 40 cycles with 15 seconds at 95˚C and 60 seconds at 60˚C. Samples were amplified in triplicates using Applied Biosystems StepOnePlus™ Real-Time PCR System (Applied Biosystems). A

melting curve was performed as a control measure for non-specific amplification. Absolute amounts in each sample were obtained from the standard curve.

### 2.11 Protein load

Bromocresol Purple (BCP) Albumin Assay Kit (Sigma-Aldrich, Merck, Germany) performed as per manufacturer's instructions with plasma samples diluted 5-fold in ultrapure water. The kit utilizes bromocresol purple, which forms a colored complex specifically with albumin. The intensity of the colour, which is directly proportional to the albumin concentration in the sample, was measured at 610 nm with a 96 well plate reader (POLARstar Omega, BMG Labtech).

### 2.12 Statistical Analysis

Statistical significance was determined by an unpaired parametric Student t-test. Unless specified, the p-value significance threshold was set at 0.05. When reporting on statistical significance symbols '*ns*' is used to indicate non-significance (p>0.05), while *, **, *** denotes p<0.05, p<0.01, p<0.001 as per conventional practice.

### 2.13 Safety Notice

To accommodate the large rotor adaptor, the original safety lid of the mini-centrifuge was removed. During the development phase, other safety measures were put into place in case of an adverse event (e.g., accidental detachment of rotor adaptor). Firstly, the base of the micro-centrifuge was fixed with a 5 mm Perspex sheet with recessed screws. Secondly, all the experiments were performed under a custom 6 mm thick Perspex safety hood. All the designs went through a 30-minute continuous runtime without incident. None of the designs showed any overheating, rotor displacement/detachment. The motor power load was measured on A-E0 designs on a Bench Digital Multimeter (Keithley DMM6500, Tektronix, Beaverton, OR, USA) (S4 Table in S1 File) and found to operate well below the indicated power rating of the original instrument at full speed. Peak power loads within 10% of the original power rating were observed at the beginning of each cycle. Motor power load measurements are available on Figshare at https://doi.org/10.6084/m9.figshare.16762444.v2. While we are confident the adapted rotor can be safely operated, we recommend additional precautions, such as (i) performing a thorough visual inspection of the printed parts to ensure they have no defects (ii) ensuring the correct balance of the tubes prior to spin, and (iii) operating the instrument within an appropriate enclosure (we recommend at least 5 mm thick solid polycarbonate).

## 3. Results and discussion

### 3.1 Design statement and design progression

The main goal of the design was to achieve the separation of plasma from blood in least one 9mL Sarstedt tube within 10 minutes, with separation performance equivalent or better than a control on a commercial centrifuge. A full set of requirements was drawn, shown in Table 1. A total of thirteen designs were manufactured and tested. Each design is presented in Fig 3 alongside physical parameters. All the designs developed were capable of holding two 9 mL S-Monovette tubes (Requirement R1) as well as all installation requirements (R2-R5). To achieve operational requirements R6-9, engineering parameters such as the size of the device, weight, radial distance, aerodynamic drag force around the device and the rotational angle were explored. Design A and B were the preliminary rotor designs and were planned without aerodynamic drag force considerations. The primary objective of these designs was to have a smaller footprint and utilise less material, lowering the end centrifugation cost to a minimal.

**Table 1. Design requirements.**

| Reference | Requirements |
|-----------|--------------|
| Capacity requirements | |
| R1 | The centrifuge must have capacity for at least two 9mL Sarstedt monovette tubes |
| Installation requirements | |
| R2 | The centrifuge must work properly under the normal range laboratory environment: Temperature 15–25 C, humidity 30–50% |
| R3 | The centrifuge must be bench top and must not move during operation |
| R4 | The centrifuge must be easy to clean |
| R5 | The centrifuge must be able to operate with electrical power |
| Operational requirements | |
| R6 | The centrifuge must have vibration-free performance, defined as hemolysis of samples not exceeding 2% |
| R7 | The maximum speed achieved must be equal (within 5%) to the speed of the original devices: 7,000rpm |
| R8 | The centrifuge must be capable of providing same or better yield and separation performance (residual plasma RBC) as control within 10 min of operation |
| Health and Safety requirement | |
| R9 | The centrifuge should be turned on and off in a safe manner, which does not bring the user in contact with moving parts |
| R10 | The noise level generated during operation of the equipment should not exceed the level of 90 dB |
| R11 | The centrifuge must provide sample tube protection |

For these two devices, We varied the angles and radial distance of these two designs while keeping the weight constant around 60 g to observe the radial distance effect on the RCF.

Although the radial distance of design A is higher than that of design B, Design A had smaller final RCF. A close inspection of angle variations also showed similar phenomena. The

| Design | Schematic | Cross section | Tube Angle (°) | Rotor mass (g) | Average radial distance, $\bar{r}$ (cm) | Max. body radius (cm) | Rotational speed (RPM) | Applied RCF (×g) |
|--------|-----------|---------------|----------------|----------------|------------------------------------------|------------------------|------------------------|------------------|
| A | | N/A | 30 | 60.7 | 7.677 | 10.1875 | 1706 | 250 |
| | | | 37.5 | 59.9 | 8.337 | 10.925 | 1607 | 241 |
| | | | | 60.5 | 7.979 | 10.55 | 1719 | 264 |
| | | | 45 | 60.33 | 8.979 | 11.55 | 1524 | 233 |
| | | | | 60.8 | 9.979 | 12.55 | 1345 | 202 |
| B | | N/A | 30 | 59.3 | 6.204 | 8.57 | 2101 | 307 |
| | | | 37.5 | 59.2 | 6.596 | 9.091 | 1990 | 293 |
| | | | 45 | 59.1 | 7.135 | 9.65 | 1909 | 291 |
| C | | | 45 | 364.8 | 4.459 | 9.25 | 4160 | 864 |
| D | | | 45 | 384 | 3.835 | 8.275 | 4505 | 871 |
| E0 | | | 25 | 297.3 | 2.278 | 6.25 | 6392 | 1043 |
| E2 | | | 25 | 338.6 | 2.094 | 6.05 | 6725 | 1060 |
| E5 | | | 25 | 272.8 | 2.066 | 5.75 | 6884 | 1097 |

**Fig 3. Evolution of design and basic characterisation.**

radial distance of designs with higher angle was slightly higher than the designs with smaller angle. As an example, the radial distance of design B-45˚ is 0.931 mm higher than that of B-30˚ design and the RCF value is 16×g lower. This effect can be explained by the aerodynamic drag force acting during the rotation which is directly related to the drag coefficient (depends on the shape of the body), the frontal area and the square of the rotating velocity [36]. The frontal area of design A and B can be compared to a flat plate that is highly resistive to the surrounding air and have a high drag coefficient of 1.17 to 1.98 [37, 38]. With a higher radial distance or angle, the frontal area becomes even larger and minimize the speed of the device and its RCF value. Thus, the maximum achievable speed of designs A and B is much smaller than the rated speed of the motor because of their high air resistive flat plate shape. Therefore, although these initial arbitrary shaped designs have some advantages such as the low material amount and low material cost, and fast printing time (<2 hours), the achieved speed and RCF are not capable of highly efficient centrifugal separation (Requirements R8&R9).

In industrial centrifugation, conical designs such as cone-stack type centrifuge [39, 40], and conical plate centrifuge [41] are prevalent. In subsequent designs (C-E), we adopted a truncated cone shape design to achieve a drag coefficient to 0.05–0.5 and thus reduce the aerodynamic drag force operating around the rotor body. A schematic diagram of the aerodynamic drag force around the designs A-B and C-E is provided in Fig 4A. The speed of the truncated conical shape C-E devices was over two folds that of the primary designs A-B. The relationship between the radial distance and speed for all designs is displayed Fig 4B.

After achieving a higher speed (4160 RPM) with the original truncated cone shape design C, speed was further increased with the later designs D-E by decreasing the lateral surface area (Fig 4C). Design with a smaller lateral surface area provided the least air resistance during the rotation which increased the speed of these designs. The final design E5, achieved a rotational speed of 6884 RPM, almost the rated speed of the original mini-centrifuge, and a RCF value of around 1100×g. The final RPM is higher than that of many previously designed centrifuges [16, 18, 19, 21, 42] which was critical in achieving our operational requirement R8. Despite the fact that some of the previously developed devices have similar [20] or higher [17] RPM, our design stands apart for severa reasons. Firstly, none of the high-speed devices can handle samples with large volumes (>9 mL). Secondly, because our design can directly accommodate this widely used Monovette blood handling tubes, there is no need to process the blood after withdrawal. Our design is simple, consisting of three parts: a tube holder, as well as a base and lid that can be threaded with the tube holder. It can be noted that the final three designs (E0, E2 and E5) all possess a 25˚ rotational angle. Although a higher angle is recommended to ensure the most compact pellet, the trade-off has to be made to increase the speed and RCF of the designs. The evaluation of all designs against the set list of requirements is available as (S5 Table in S1 File).

## 3.2 Truncated cone shape design and vibrational analysis

In the previous section, we discussed that by considering the aerodynamic drag force with the first truncated cone shape design, Design C, the rotational speed was increased almost 2-fold compared to design B where a lower speed has been observed because of its flat plat like structure which increased the aerodynamic drag force of the body. However, with higher speeds, vibrations were observed during the ramping up or slowing down of the rotation. In order to understand the nature of these vibrations and reduce them, we used video analysis to measure their amplitude and duration (see Material and Methods section). Fig 5A illustrates this process on Design C. Fig 5Ai and 5Aii show snapshots of the recorded video before, and after, the edge detection with the Python OpenCV code. Fig 5Aiii shows the deflection measurements

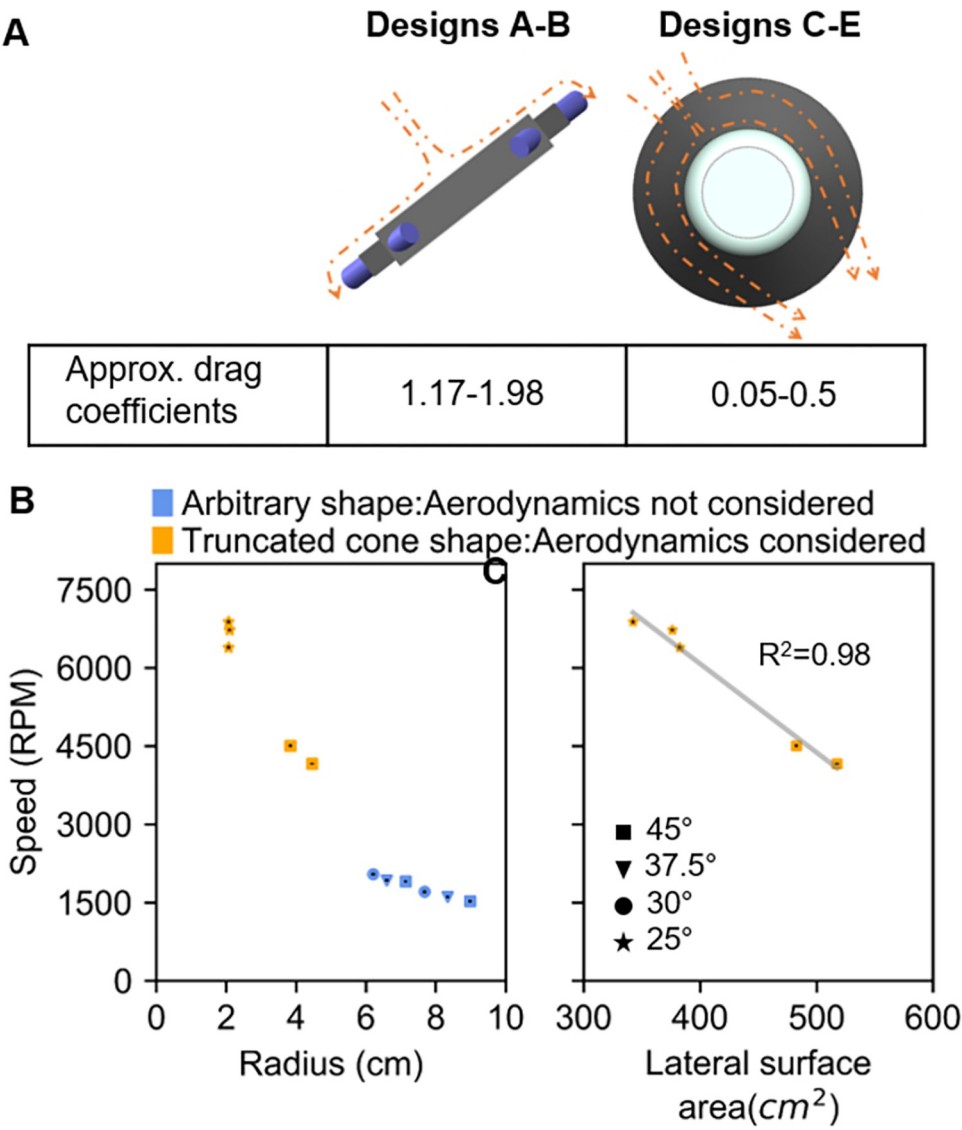

**Fig 4.** Influence of design shapes and sizes on speed **(A)** Schematic diagram showing an arbitrary shape design provides higher aerodynamic drag force than the truncated cone shape design by resisting much of the surrounding airflow. For simplicity, airflow is showed only in one direction instead of all sides. **(B)** Radial distance of all designs vs their rotational speed **(C)** Lateral surface area of the truncated cone-shaped designs vs their rotational speed. NB: Both the panel B and C share the same legend.

obtained from one of the Design C videos. A deflection around 4 mm was observed both in the accelerating and decelerating phase close to 800 RPM, the critical speed of Design C. These vibrations at high speed could damage the RBCs because of high shear stress, hence the primary target was to lower the critical speed of the conical shape designs while ensuring higher rotational speed. To achieve this purpose, the lateral surface area of the abovementioned designed devices was decreased to boost the rotational speed. To predict the critical speed before printing the devices, we used in-silico Ansys Finite Element Analysis (FEA).

The fine meshing of design C and the mode shape during the deflection of the device can be observed from Fig 5Bi and 5Bii, for illustration. The Ansys model correctly predicted the critical speed of each model with high accuracy (model vs experimental measurement values

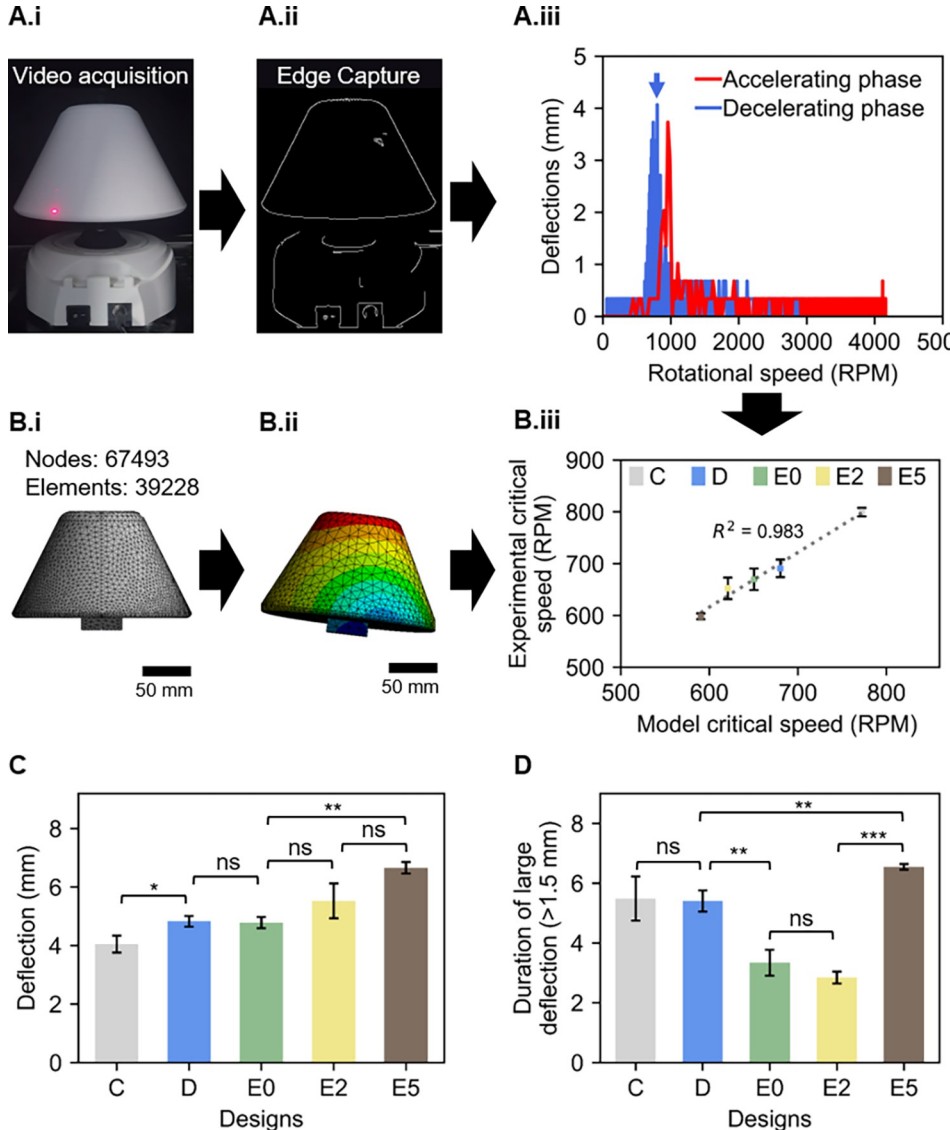

**Fig 5.** (A) Deflection measurement from the recorded video of conical Design C. **A.i)** Snapshot of one of the video recordings of Design C **A.ii)** Edge capturing from one of Design C videos using Canny, an edge detection operator in Python OpenCV module. Further details about the deflection measurements are available in (S1 Fig in S1 File) **A.iii)** Measured deflection of both acceleration and decelerating phase from the recorded video using python OpenCV module for Design C. All recordings are available from (S2 Fig in S1 File) **(B)** Critical speed measurement in Ansys workbench **B.i)** Meshing of Design C (Nodes: 6749, Elements: 39228) C **B.ii)** Mode shape during natural frequency **B. iii)** Simulated (model) vs experimental critical speed of each conical shape design showing a good agreement between them with low percentage difference (1–5%) **(C)** Measured deflection of each conical shape design showing all designs experiencing deflection from 4 to 7 mm **(D)** Duration of deflection higher than 1.5 mm was decreasing the lowering of critical speed except design E5 which experienced higher deflection for longer times.

in Fig 5Biii, $R^2$ = 0.983). The maximum deflection of each design at their critical speed can be found in Fig 5C. The final design, E5 with the lowest critical speed of around 600 RPM provided a maximum deflection of around 7 mm whereas for the other designs the average deflection was limited within 4 to 5 mm. These deflections can be interpreted as impulsive vibrations which might exert excessive shear stress on the RBCs and cause hemolysis [43, 44]. Therefore, the final degree of hemolysis depends on the shear stress exerted on RBCs and the

exposure time of the impulse vibration [45]. Fig 5D illustrates the duration of deflection higher than 1.5 mm. The duration of large deflections was minimised from C to E2. However, design E5 appeared to be providing maximum deflection for about 7 seconds. Therefore, despite having the highest rotational speed and lowest critical speed, we predicted that E5 would have a low separation efficiency because of the amplitude and the duration of these impulsive vibrations. This was verified during the biological characterisation of the devices.

### 3.3 Plasma yield

Following the physical characterisation of the devices, we proceeded to the biological characterisation of Designs C-D to investigate if the requirement R8 was met. the first, we investigated the separation yield, which indicates how much plasma volume the centrifuge can separate from the total volume of available plasma. To determine the plasma separation yield, 9 mL of pooled blood samples were centrifuged with each conical design (C, D, E0-2-5) and the full–scale centrifuge control for 3, 6 and 10 minutes and the separated plasma volume and full blood count were measured. Fig 6 shows photographs (panel A) and quantitative results (panel B) of plasma yield in the initial sample and samples after 3, 6, and 10 minutes centrifugation on the adapted centrifuge and the control.

It is worth noting that most of the plasma gets separated within 3 minutes of centrifugation (minimum separation ~70% for Design C) and the separation volume further increases at 6 and 10 minutes. The single most striking observation is the decreased separation performance of Design E5 at 6 and 10 minutes. Device E5 shows much greater performance after 3 minutes centrifugation (~90% yield) compared to the control. However, during 6- and 10-minutes separation instances, the design failed to maintain this higher performance where it was unable to

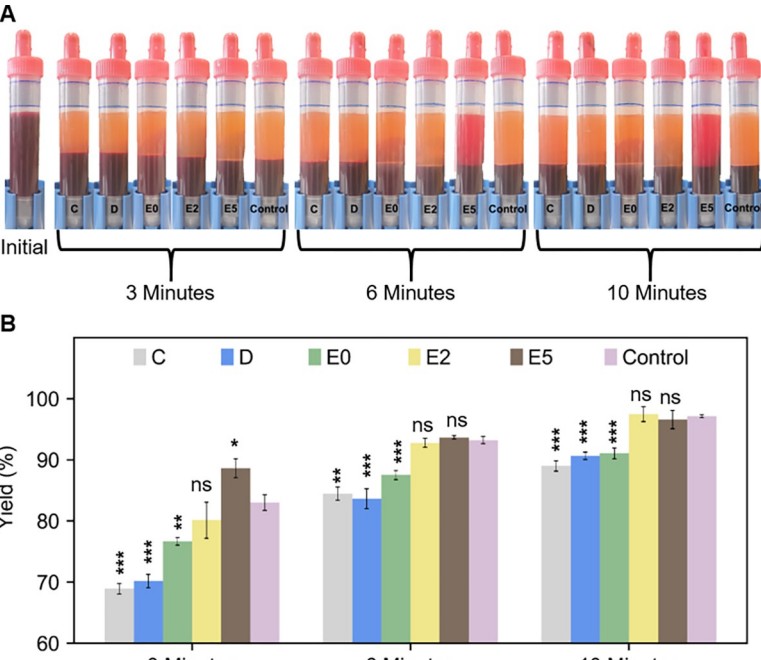

**Fig 6. (A)** Images of the initial and post centrifugated (3,6,10 minutes) S-Monovette tubes showing that larger centrifugation times resulted in a higher volume of plasma separated from 9 mL sample. **(B)** Measured separation yield of different designs compared with the control centrifugation performance. Statistics: standard unpaired t-test between each column and the control. Design E2 and E5 were able to separate the same amount of plasma as the control at 6 and 10 minutes with no statistically significant difference.

separate all RBCs present in the separated plasma. To facilitate the discussion here, it needs to be clarified that this reddish colour of plasma at 6 and 10 minutes was exclusively because of the nonseparated red blood cells present in plasma, not due to any RBC damage which was quantified and will be described later. The internal shape of design E5 might be a reason behind this poor separation efficiency. This design is comparatively the smallest of all the other designs and to accommodate this size, the sidewall was brought very close to the placed tube inside and furthermore, the tube was completely encased against the sidewall, unlike the other design where the tubes are hanging freely, away from sidewalls. In the previous section, we noticed a high deflection and duration of deflection with design E5 which we suspect might cause the sidewall to touch the S-Monovette tubes randomly at several instances and hinder the separation efficiency. With device E2, it was possible to achieve almost 80% yield within 3 minutes which was almost equivalent to the performance of the control. The comparative yield of design E2, E5 (approximatively 95%) are not significantly different from the control after 10 minutes of centrifugation.

As discussed previously, the overarching goal in this study was to customize a low-cost microcentrifuge in such a way that it could handle a large 9 mL clinical sample volume directly after the blood withdrawal without any extra blood handling steps. At the same time, the device should ensure at least similar separation performances as a benchtop refrigerated centrifuge. We, and others in the field, have previously demonstrated microfluidic solutions as an alternative to traditional centrifugation [46–49]. However, we have concluded that for extremely low biomarker detection such as the detection of various fractions of circulating DNA, microfluidic approaches are not viable because of their low separation yield and the difficulty to handle the large volume of the viscous whole blood sample. The main barrier in the microfluidic approach is the cell-cell interaction between red blood cells. When high volume fractional blood is flowed at a high flow rate, the cell interaction between a large number of RBCs in the microfluidic channel increases, which inhibits the cell-free layer formation and other deterministic effects, thus reducing the separation performance [50]. As a result, the use of high-volume fractional blood in the above comparative high-throughput studies has resulted in much lower separation yields. Comparatively, we can see in this study, design E0 and E2 managed to secure more than 95% of the available plasma within 10 minutes of centrifugation. This high plasma yield is a valuable factor in low-level biomarker detection.

## 3.4 Plasma quality: Residual blood cell count

Firstly, the total RBC count was measured each time the blood sample was remixed following centrifugation to investigate the integrity of the blood sample after centrifugation. Fig 7A presents the pre-and post-centrifuged whole blood RBC count at 3, 6 and 10 minutes. It is apparent that there was no significant decrease in RBC count between centrifugations. Due to the removal of plasma (the hematology analyser removes around 50 μL of the sample during each measurement) at every time instance, we noticed a slight increment in the RBC count per litre of blood on some of the designs (and control). The absence of hemolysis in most of the samples (see Section 3.6) corroborate this interpretation.

Secondly, a residual cell count can be used to establish the quality of the separated plasma. Fig 7B shows the remaining RBC count on the separated plasma. The remaining RBCs in design E2 was significantly lower than that of the control at all time instances. Design E5 performed best at 3 minutes, however, due to the vibrations reported earlier in this work, the separation reversed at 6 minutes. The best cell separation performance was observed in the platelet counts (Fig 7C). All adaptors (apart from E5) were capable of separating more platelets than the control because of their much higher speed. Notably, design E2 separated almost 4

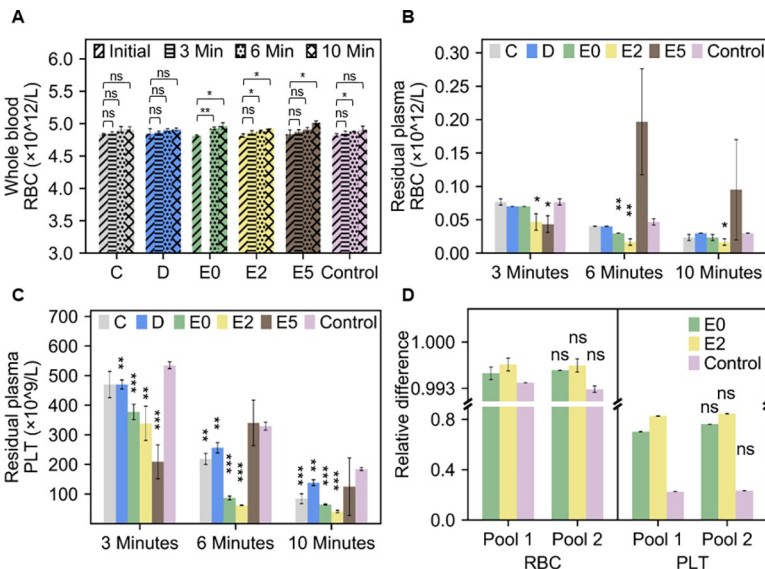

**Fig 7.** Blood count at 3, 6 and 10 minutes **(A)** The red blood cell counts of pre (initial) and post (3,6, 10 minutes) separated blood show the integrity of the sample after centrifugation. The statistical indications relate to a non-paired t-test between the RBC counts between 0, 3, 6, and 10 minutes **(B)** The red blood cell (RBC) count on plasma after centrifugation shows RBC concentration with design E2 was significantly lower than the control in all periods. On the contrary, design E5, although performing well at 3 minutes was hampered by vibrations reversing the separation process. Here the statistics relate to a non-paired t-test between each of the designs and the control, for each time point **(C)** Platelet count (PLT) on separated plasma. Design E2 was able to separate almost 4 times higher platelet than the control within 10 minutes. Here the statistics relate to a non-paired t-test between each of the designs and the control for each time point **(D)** Relative RBC and Platelet counts after 10 minutes centrifugation from two different pools of blood. Here the statistics relate to a non-paired t-test between Pool 1 and Pool 2 for each experiment. No significant difference was observed between different pools of blood.

times more platelets than the control, which could be of benefit in coagulation studies requiring platelet-poor plasma. As explained in Materials and Methods, the separation performance was investigated on pooled blood samples (see Materials and Method section), which has the advantage of removing individual sample specificities and enables accurate comparison between designs and control. To investigate any notable differences between individual pools, we compared the separation performance of designs E0, E2 and control after 10 minutes of centrifugation with two different pools of blood (Fig 7D). The two pools investigated were found to have significant differences in original Hct (Pool 1, Hct $\approx$ 45.5%; Pool 2, Hct $\approx$ 43.2%). However, in terms of relative difference post and pre-centrifugation, the pools showed no significant differences in RBC and platelet relative counts.

### 3.5 Plasma quality: Hemolysis detection

To assess the quality of the separated samples, we measured the free-hemoglobin released after centrifugation and derived a hemolysis percentage (See Materials and Methods). Hgb levels from collected plasma samples of design E0, E2 and control (single and double centrifugation) were measured using Cripps method (See Materials and Methods). As reported in Fig 8A, all the measured samples had an estimated percentage of hemolysis between 1 to 1.7% corresponding to a free Hgb concentration of 0.2–0.4 g/dL. Despite the deflections observed on Designs E0 and E2, the separated plasma is well below the threshold of hemolysis limit and similar to the control used in our experiment, which shows the adapted device is adequate for general low-speed spin of clinical blood samples.

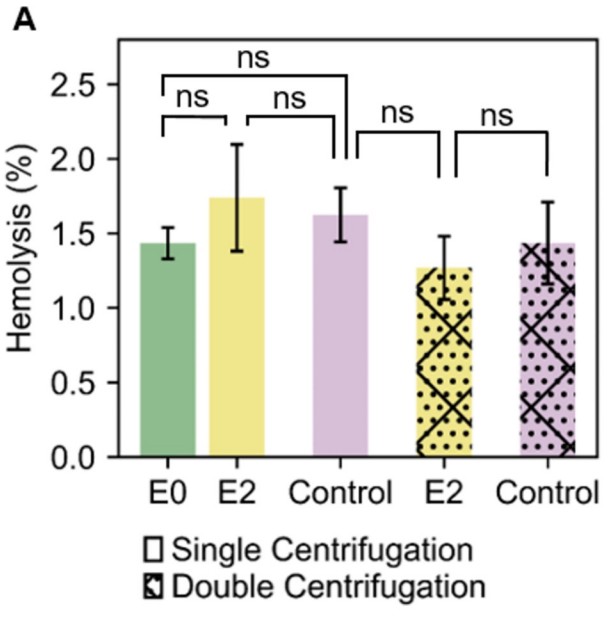

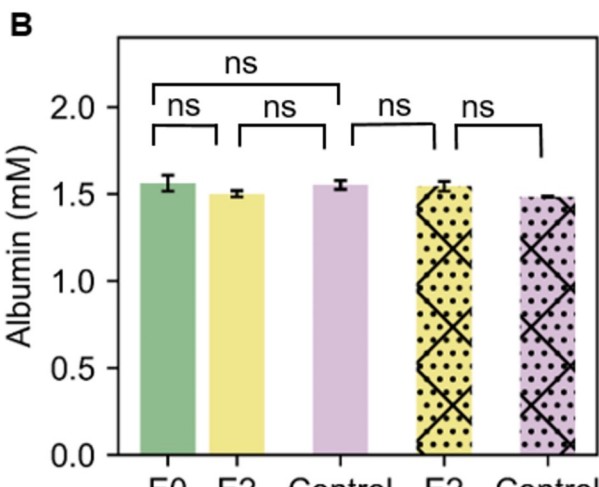

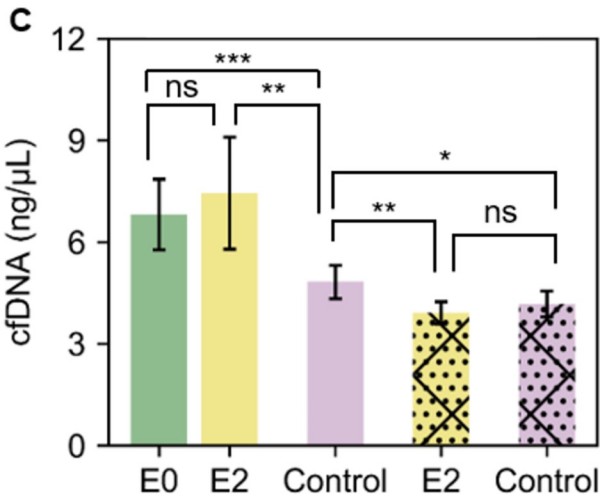

**Fig 8. (A)** Percentage of hemolysis in separated plasma from E0, E2 and control. **(B)** Albumin concentration in E0, E2 and control **(C)** Cell-free DNA levels in E0, E2, and control after single centrifugation or with additional second separate centrifugation.

## 3.6 Plasma quality: Protein load

Albumin is known to be the most abundant plasma protein in humans which accounts for almost 60% of the total serum protein. It is an established biomarker of nutritional status [51] and reliable prognostic indicator for heart failure [52], pulmonary arterial hypertension [53], in acute surgical patients [54], morbidity and mortality [55] and many more diseases. The recovered plasma from the best designs (E0, and E2) and control centrifugation (single and double) were analysed to quantify the concentration of albumin. The albumin quantification from the sample was obtained by directly measuring the absorbance at 610 nm when the available albumin in a sample provides a peak in the presence of reagent bromocresol purple (see Materials and Methods). As shown in Fig 8B, E0 and E2 samples (single and double centrifugation) showed a similar concentration to the full-scale centrifuge control, around 1.5 mM (~2.5 g/dL) with no statistically significant difference which shows the adapted centrifuge is suitable for plasma protein studies. Similar albumin level has been detected in previous studies [49]. The albumin concentration in all samples, including control samples, was slightly lower than the normal range of Albumin in plasma (3.4 to 5.4 g/dL), which can be explained by the age of the samples (3 days old).

## 3.7 Plasma quality: cfDNA extraction

Cell-free DNA (cfDNA) are small (50-200bp) DNA fragments that originate from cell apoptosis or necrosis. cfDNA overall levels or specific cfDNA regions can be used biomarkers in the diagnosis, prognosis, or monitoring of fetal chromosomal abnormalities, various cancers, infectious diseases and organ transplants [56]. This is an emerging biomarker that is now routinely and increasingly used in clinical practice. To assess the suitability of the adapted centrifuge for cfDNA-based assays, we measured the levels of cfDNA using a LINE PCR. Total cfDNA was extracted from 3 mL of separated plasma as per Material and Methods. After the first centrifugation, we found the cfDNA concentration from design E0 and E2 to be higher (6.5–8 ng/μL) compared to the control (~5 ng/μL) (Fig 8C). This could be due to the substandard pelleting of white blood cells singe significantly more residual white blood cells (WBCs) were found in the adapted centrifugation, compared to the control (S3 Fig in S1 File). The lysis of remaining cells (including WBCs) during the cfDNA extraction has introduced genomic DNA, resulting in overall higher cfDNA levels. Other studies also have reported higher cfDNA concentration after single centrifugation [27]. A second, higher speed centrifugation is often incorporated in cfDNA extraction protocols. This second spin at a higher speed (12,000×g RCF for 10 minutes), can be performed on smaller sample volumes using cheaper and smaller bench centrifuge. To test if cfDNA yields can be brought to the same level as the control, we applied a second spin to both design E2 and control samples. After this second centrifugation, both sample types showed a similar lower concentration of cfDNA with no statistically significant difference. Generally speaking, it can be noted that the total cfDNA levels are higher than levels reported elsewhere, this is due to age of the samples (three days old) which is sub-optimal for cfDNA-based diagnostic, but adequate to characterise the performance of a device. Although designs E0 and E2 showed similar RBCs separation and the yield compared to that of control, the low rotational angle of 25° resulted in lower buffy coat compaction and consequently higher WBC counts in the separated plasma, compared to the benchmark. This issue might be addressed by exploring the use of a higher rotational angle or longer spin duration.

In the meantime, a second high-speed centrifugation can remove remaining WBCs and lead to similar cfDNA levels in the adapted design and benchmark.

## 4. Conclusion

In this study, a mini-centrifuge costing <$130 was adapted to handle large volume (9 mL) standard clinical samples in S-Monovette collection tubes. Our results showed that the air resistance was the most crucial parameter needed to be optimised to ensure an adequate centrifugal force for comparable cell separation to a commercial centrifuge. Following optimisation, the final design reached a speed of around 6725 RPM and RCF of 1060×g. Similar yield, cell counts (red blood cells), hemolysis and albumin levels were obtained from the optimised design and benchmark. Furthermore, the performance of this customised centrifuge in platelet separation is better than the control due to its high speed, which can be useful in coagulation studies. Only the total cfDNA levels of the plasma separated in the adapted design were found to be significantly different to the benchmark, owing to a reduced separation performance for white blood cells. This can be alleviated by further optimisation of vibration or potentially by a slightly longer spin time. The nearest cheapest option for 9mL S-Monovette centrifugation cost over $1,500. The bill of material for our adapted centrifuge stands at around $140. The overall performance of the optimised adapted centrifuge, which costs a fraction of the total price of the commercial control centrifuge, was equivalent to a commercial centrifuge and superior to microfluidic approaches in yield, throughput and quality. Therefore, our design offers value and performance to the low-resource environment or could be further adapted to created portable diagnostic laboratories.

## Supporting information

**S1 File. Supplementary information.**
(DOCX)

**S1 Code. Python code for deflection measurements.**
(DOCX)

**S1 Video. Video recording of Design C.**
(MP4)

## Author Contributions

**Conceptualization:** Md Ehtashamul Haque, Alvaro J. Conde.

**Data curation:** Md Ehtashamul Haque, Linda Marriott.

**Formal analysis:** Md Ehtashamul Haque, Maïwenn Kersaudy-Kerhoas.

**Funding acquisition:** Maïwenn Kersaudy-Kerhoas.

**Investigation:** Md Ehtashamul Haque, Taygan Henry, Maïwenn Kersaudy-Kerhoas.

**Methodology:** Md Ehtashamul Haque, Linda Marriott, Noman Naeem, Alvaro J. Conde, Maïwenn Kersaudy-Kerhoas.

**Project administration:** Md Ehtashamul Haque, Maïwenn Kersaudy-Kerhoas.

**Resources:** Maïwenn Kersaudy-Kerhoas.

**Software:** Md Ehtashamul Haque.

**Supervision:** Alvaro J. Conde, Maïwenn Kersaudy-Kerhoas.

**Validation:** Md Ehtashamul Haque.

**Visualization:** Md Ehtashamul Haque, Maïwenn Kersaudy-Kerhoas.

**Writing – original draft:** Md Ehtashamul Haque, Maïwenn Kersaudy-Kerhoas.

**Writing – review & editing:** Linda Marriott, Noman Naeem, Alvaro J. Conde.

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
