## [Decision Letter · Decision Letter 0]

14 Jan 2022

PONE-D-21-33309Design, 3D-printing, and characterisation of a low-cost, open-source centrifuge adaptor for separating large volume clinical blood samplesPLOS ONE

Dear Dr. Kersaudy-Kerhoas,

Thank you for submitting your manuscript to PLOS ONE. After careful consideration, we feel that it has merit but does not fully meet PLOS ONE’s publication criteria as it currently stands. Therefore, we invite you to submit a revised version of the manuscript that addresses the points raised during the review process.

We look forward to receiving your revised manuscript.

Kind regards,

Denni Kurniawan

Academic Editor

PLOS ONE

Journal Requirements:

Additional Editor Comments:

Please observe the comments from Reviewers. The first Reviewer commented on the presentation style and the second Reviewer suggested some clarifications. Please respond with your response which will also be reflected in the manuscript or rebuttal.

Reviewers' comments:

Reviewer's Responses to Questions

**Comments to the Author**

1. Is the manuscript technically sound, and do the data support the conclusions?

Reviewer #1: Yes

Reviewer #2: Yes

2. Has the statistical analysis been performed appropriately and rigorously? 

Reviewer #1: Yes

Reviewer #2: Yes

3. Have the authors made all data underlying the findings in their manuscript fully available?

Reviewer #1: Yes

Reviewer #2: Yes

4. Is the manuscript presented in an intelligible fashion and written in standard English?

Reviewer #1: Yes

Reviewer #2: Yes

5. Review Comments to the Author

Reviewer #1: 1. Title is too long and not fully reflect to the work done

2. The abstract is not clear and precise. Author should include some details :

a. Re-establish the topic of the research.

b. Give the research problem and/or main objective of the research (this usually comes first).

c. Indicate the methodology used.

d. Present the main findings.

e. Present the main conclusions

3. The introduction section not has a clear statement demonstrating that the focus of the study. The problem definition need to state clearly. There also a need to give a brief, well-articulated summary of research literature that substantiates the study. The paper introduction should shows:

a. The importance of research

b. The benefit of research

c. People who will get the benefit of the research

d. Area which need to be improved

e. The gap that will be filled by this research

4. The purposes, research question(s), and/or hypotheses appropriate to the topic and area of the study are not well explained. Can add details to enhance reader understanding

5. The paper mentions the literature review/related works. Author should include these in literature review.

a. Show appropriate preparation and knowledge through the background/review of literature in the related works.

b. The paper shows how it relates to the other work.

6. Author need to mention how data was collected in order to help the reader to evaluate the validity and reliability of the results, and the conclusions draw from them.

7. The paper has lack of explanation of how data was collected/generated, explanation of how data was analyzed explanation of methodological problems and their solutions or effects

8. Results are presented clearly and analyzed appropriately. However author can improve by cover:

a. Statement of results: the results are presented in a format that is accessible to the reader.

b. Tables and figures should be accompanied by text that guides the reader's attention to significant results

9. Discussion is not presented clearly and analysed appropriately. Author may add reference from previous research as justification and comparison.

10. The conclusions can be improved by giving adequately summary of:

a. What was learned

b. What remains to be learned (directions for future research)

c. The shortcomings of what was done (evaluation)

d. The benefits, advantages, applications of the research (evaluation), and recommendations.

11. Some of references are not up to date. Please used latest 5 years back.

Reviewer #2: This is an interesting topic and the manuscript is well-written. The authors have clarified several issues and took them seriously during the design consideration and test experiments. However, some suggestions to consider are as following:

The clarification of design concepts is limited.

The results may be presented in a more understandable format for readers.

Figure formatting is inconsistent i.e. 76 - Figure 1: A), 90 – Figure 2: (A), etc.

88 – should be w not w2

For Deflection measurement, no reason stated for using a duration of 4 minutes.

Overall, I have no objection to this manuscript being published.

6. PLOS authors have the option to publish the peer review history of their article (what does this mean?). If published, this will include your full peer review and any attached files.

Reviewer #1: No

Reviewer #2: No

---

## [Author Response · Author response to Decision Letter 0]

7 Mar 2022

//A formatted version of this response is available from uploaded documents

Answers to Reviewers:

We would like to thank the reviewers for their time and their general appreciation of our manuscript. Based on their reviews, we have modified text and figures. Our responses to the individual comments are detailed below (blue). We have uploaded an annotated revised manuscript and a clean revised manuscript. 

The authors

Referee: 1

Comment 1. Title is too long and not fully reflect to the work done

Authors: We propose the following shorter title as “A low-cost, open-source centrifuge adaptor for separating large volume clinical blood samples” and hope this is satisfactory. We would be most willing to receive other suggestions from the reviewers or editors.

Comment 2. The abstract is not clear and precise. Author should include some details:

a. Re-establish the topic of the research.

b. Give the research problem and/or main objective of the research (this usually comes first).

c. Indicate the methodology used.

d. Present the main findings.

e. Present the main conclusions

Authors: We have revised the abstract, taking into account the reviewer comments. The revised abstract is available in the revised manuscript (Tracked change and clean document appended). 

This now reads:

Blood plasma separation is a prerequisite in numerous biomedical assays involving low abundance plasma-borne biomarkers and thus is the fundamental step before many bioanalytical steps. High-capacity refrigerated centrifuges, which have the advantage of handling large volumes of blood samples, are widely utilized in this regard, but they are bulky, non-transportable, and prohibitively expensive for low-resource settings, with prices starting at $1,500. On the other hand, there are low-cost commercial and open-source micro-centrifuges available, but they are incapable of handling large amounts of blood samples. There is currently no low-cost CE marked centrifuge that can process large volumes of clinical blood samples on the market. As a solution, we customised the rotor of a commercially available low-cost micro-centrifuge (~$125) using 3D printing to enable centrifugation of large clinical blood samples in resource poor-settings. Our custom adaptor ($15) can hold two 9 mL S-Monovette tubes and achieve the same separation performance (yield, cell count, hemolysis, albumin levels) as the control benchtop refrigerated centrifuge, and even outperformed the control in platelet separation by at least four times. This low-cost open-source centrifugation system capable of processing clinical blood tubes could be valuable to low-funded laboratories or low-resource settings where centrifugation is required immediately after blood withdrawal for further testing.

Comment 3. The introduction section not has a clear statement demonstrating that the focus of the study. The problem definition need to state clearly. There also a need to give a brief, well-articulated summary of research literature that substantiates the study. The paper introduction should shows:

a. The importance of research

b. The benefit of research

c. People who will get the benefit of the research

d. Area which need to be improved

e. The gap that will be filled by this research

Authors: We have significantly revised the introduction, taking into account the reviewer comments. The revised introduction is too long to paste here, but is available in the revised manuscript (page 2-3 in Tracked change version).

Comment 4. The purposes, research question(s), and/or hypotheses appropriate to the topic and area of the study are not well explained. Can add details to enhance reader understanding

Authors: Taking example of another PLOS One paper on an engineering subject, and the reviewer’s comments, we have however brought some modifications, which we hope will satisfy the reviewer. We have extensively changed the introduction and included a clear design statement in the newly named Section 3.1. “Design statement and design progression” and a new table specifying a full list of requirement (New Table 1) and a Supplementary table with a review of each design against that list of requirement (Suppl Table 5). 

Comment 5. The paper mentions the literature review/related works. Author should include these in literature review.

a. Show appropriate preparation and knowledge through the background/review of literature in the related works.

b. The paper shows how it relates to the other work.

Authors: a) We have added the following reference: Bhupathi, Chinna, and Devarapu 2021; Brown et al. 2011; Li et al. 2020; Michael et al. 2020; WareJoncas, Stewart, and Giannini 2018; Wong et al. 2008. To be the best of our knowledge we have added all relevant literature. If the reviewer has specific example we may have forgotten, or that we are not aware of, we respectfully ask that they pass these on to us for inclusion b) We have added a number of sentences to clarify how our results relate to previous studies (page 13, Track change version)

Comment 6. Author need to mention how data was collected in order to help the reader to evaluate the validity and reliability of the results, and the conclusions draw from them.

Authors: We believe we have amply described how data was collected. In addition, we have made all our data publicly accessible. If the reviewer has concerns about a specific aspect of the data collection, we would be very happy to look into it and improve. 

Comment 7. The paper has lack of explanation of how data was collected/generated, explanation of how data was analyzed explanation of methodological problems and their solutions or effects

Authors: We have added further details to Materials and Method section, Section 2.4: Simulation to critical (page 6-7, track change version) and Section 2.5: Deflection measurements (page 7, track change version). We believe we have otherwise amply described how data was collected. In addition, we have made all our data publicly accessible. If the reviewer has concerns about a specific aspect of the data collection, we would be very happy to look into it and improve. 

Comment 8. Results are presented clearly and analyzed appropriately. However, author can improve by cover:

a. Statement of results: the results are presented in a format that is accessible to the reader.

b. Tables and figures should be accompanied by text that guides the reader's attention to significant results

Authors: a) To the best of our ability our results are presented in an accessible format to the reader, however we would be very happy to address in further specific issues. b) All our Tables and Figures are accompanied by rich text which guides the reader’s attention to significant results. With due respect, we think this comment does not apply to our manuscript. We would be happy to address specific issues. 

Comment 9. Discussion is not presented clearly and analysed appropriately. Author may add reference from previous research as justification and comparison.

Authors: We have added a number of new references See comment 5, revised introduction on page 2-3 and we have added further comparison to previous studies Section 3.1 last paragraph, and Section 3.5, protein levels. We hope this satisfies the reviewer’s expectations. 

Comment 10. The conclusions can be improved by giving adequately summary of:

a. What was learned

b. What remains to be learned (directions for future research)

c. The shortcomings of what was done (evaluation)

d. The benefits, advantages, applications of the research (evaluation), and recommendations.

Authors: We have further modified the conclusion on Page 20. In addition, we have added a list of requirements and an evaluation of all designs against this list (New Table 1 and Suppl Table 5). We hope this will enhance the article. 

Comment 11. Some of references are not up to date. Please used latest 5 years back.

Authors: We have carefully checked all references, and we are unsure which ones the reviewer is referring to. We have added some references (See comment 5, and introduction page 2-3) to answer other points made by the reviewers. We would be most willing to address any further concerns. 

Reviewer #2: This is an interesting topic and the manuscript is well-written. The authors have clarified several issues and took them seriously during the design consideration and test experiments. However, some suggestions to consider are as following:

Comment 1: The clarification of design concepts is limited.

Authors: We have added some a clear design statement (Section 3.1, page 9), a full requirement table (New Table 1) to clarify the presentation of the design concept, and finally an evaluation table (Suppl Table 5). We hope this satisfies the reviewer’s expectations. 

Comment 2: The results may be presented in a more understandable format for readers.

Authors: We have tried to as clear as possible, but would be most willing to address specific concerns for the result formatting. As said earlier, we have added a full list of design requirements, and design evaluation.

Comment 3: Figure formatting is inconsistent i.e. 76 - Figure 1: A), 90 – Figure 2: (A), etc. 88 – should be w not w2

Authors: Thank you for bringing this up to our attention, we have corrected these points. 

Comment 4: For Deflection measurement, no reason stated for using a duration of 4 minutes.

Authors: Thank you, we have taken this into account and added a sentence to explain this choice, page 6, Section 2.5.

---

## [Decision Letter · Decision Letter 1]

28 Mar 2022

A low-cost, open-source centrifuge adaptor for separating large volume clinical blood samples

PONE-D-21-33309R1

Dear Dr. Kersaudy-Kerhoas,

We’re pleased to inform you that your manuscript has been judged scientifically suitable for publication and will be formally accepted for publication once it meets all outstanding technical requirements.

Kind regards,

Denni Kurniawan

Academic Editor

PLOS ONE

Additional Editor Comments (optional):

Reviewers' comments:

Reviewer's Responses to Questions

**Comments to the Author**

1. If the authors have adequately addressed your comments raised in a previous round of review and you feel that this manuscript is now acceptable for publication, you may indicate that here to bypass the “Comments to the Author” section, enter your conflict of interest statement in the “Confidential to Editor” section, and submit your "Accept" recommendation.

Reviewer #1: All comments have been addressed

Reviewer #2: All comments have been addressed

2. Is the manuscript technically sound, and do the data support the conclusions?

Reviewer #1: Yes

Reviewer #2: Yes

3. Has the statistical analysis been performed appropriately and rigorously? 

Reviewer #1: Yes

Reviewer #2: Yes

4. Have the authors made all data underlying the findings in their manuscript fully available?

Reviewer #1: Yes

Reviewer #2: Yes

5. Is the manuscript presented in an intelligible fashion and written in standard English?

Reviewer #1: Yes

Reviewer #2: Yes

6. Review Comments to the Author

Reviewer #1: All the highlighted comments has been answered well by author. This manuscript can be accepted accordingly.

Reviewer #2: This is an interesting study and the paper is generally well written and structured. The revisions have been amended. The authors have sufficiently improved their paper. Thank you.

7. PLOS authors have the option to publish the peer review history of their article (what does this mean?). If published, this will include your full peer review and any attached files.

Reviewer #1: No

Reviewer #2: No

---

## [Editor Report · Acceptance letter]

14 Apr 2022

PONE-D-21-33309R1 

A low-cost, open-source centrifuge adaptor for separating large volume clinical blood samples 

Dear Dr. Kersaudy-Kerhoas:

I'm pleased to inform you that your manuscript has been deemed suitable for publication in PLOS ONE. Congratulations! Your manuscript is now with our production department. 

Kind regards, 

on behalf of

Dr. Denni Kurniawan 

Academic Editor

PLOS ONE